# Advancing Global Health through Environmental and Public Health Tracking

**DOI:** 10.3390/ijerph17061976

**Published:** 2020-03-17

**Authors:** Paolo Lauriola, Helen Crabbe, Behrooz Behbod, Fuyuen Yip, Sylvia Medina, Jan C. Semenza, Sotiris Vardoulakis, Dan Kass, Ariana Zeka, Irma Khonelidze, Matthew Ashworth, Kees de Hoogh, Xiaoming Shi, Brigit Staatsen, Lisbeth E. Knudsen, Tony Fletcher, Danny Houthuijs, Giovanni S. Leonardi

**Affiliations:** 1National Research Council, Institute of Clinical Physiology, Unit of Environmental Epidemiology and Disease Registries, 56124 Pisa, Italy; 2Centre for Radiation, Chemical and Environmental Hazards, Public Health England, Didcot, Oxon OX11 0RQ, UK; helen.crabbe@phe.gov.uk (H.C.); Tony.fletcher@phe.gov.uk (T.F.); Giovanni.leonardi@phe.gov.uk (G.S.L.); 3Centre for Medical Education, Cardiff University, United Kingdom, Cardiff CF14 4XW, UK; 4Centers for Disease Control and Prevention, Atlanta, GA 30341, USA; fay1@cdc.gov; 5Direction of Environmental and Occupational Health, Santé Publique France, 94415 Saint Maurice, France; Sylvia.MEDINA@santepubliquefrance.fr; 6Scientific Assessment Section, European Centre for Disease Prevention and Control, 169 73 Solna, Sweden, Sweden; JanC.Semenza@ecdc.europa.eu; 7National Centre for Epidemiology and Population Health, Research School of Population Health, Australian National University, Canberra 2601, Australia; sotiris.vardoulakis@anu.edu.au; 8Vital Strategies, New York, NY 10005, USA; dkass@vitalstrategies.org; 9Environmental Health and Epidemiology, Brunel University, London UB8 3PH, UK; Ariana.Zeka@brunel.ac.uk; 10Medical University Centre “Nene Teresa”, Rruga e Dibres, #370 Tirana, Albania; 11National Center for Disease Control and Public Health, 0198 Tbilis, Georgia; i.khonelidze@ncdc.ge; 12Institute of Environmental Science and Research Limited, Kenepuru, Porirua 5240c, New Zealand; Matthew.Ashworth@esr.cri.nz; 13Swiss Tropical and Public health Institute, Basel, Switzerland, 4051 Basel, Switzerland; c.dehoogh@swisstph.ch; 14University of Basel, Basel, 4001 Basel, Switzerland; 15National Institute of Environmental Health, Chinese Center for Disease Control and Prevention, Beijing 100021, China; shixm@chinacdc.cn; 16National Institute for Public Health and the Environment, 3720BA Bilthoven, The Netherlands; Brigit.staatsen@rivm.nl (B.S.); Danny.Houthuijs@rivm.nl (D.H.); 17Department of Public Health, Denmark University of Copenhagen, 1353 Copenhagen, Denmark; liek@sund.ku.dk; 18London School of Hygiene and Tropical Medicine, London WC1E 7HT, UK

**Keywords:** environmental health, public health, global health, environmental public health tracking, surveillance, hazard, exposure and health outcomes, environmental epidemiology, health policy, prevention strategy

## Abstract

Global environmental change has degraded ecosystems. Challenges such as climate change, resource depletion (with its huge implications for human health and wellbeing), and persistent social inequalities in health have been identified as global public health issues with implications for both communicable and noncommunicable diseases. This contributes to pressure on healthcare systems, as well as societal systems that affect health. A novel strategy to tackle these multiple, interacting and interdependent drivers of change is required to protect the population’s health. Public health professionals have found that building strong, enduring interdisciplinary partnerships across disciplines can address environment and health complexities, and that developing Environmental and Public Health Tracking (EPHT) systems has been an effective tool. EPHT aims to merge, integrate, analyse and interpret environmental hazards, exposure and health data. In this article, we explain that public health decision-makers can use EPHT insights to drive public health actions, reduce exposure and prevent the occurrence of disease more precisely in efficient and cost-effective ways. An international network exists for practitioners and researchers to monitor and use environmental health intelligence, and to support countries and local areas toward sustainable and healthy development. A global network of EPHT programs and professionals has the potential to advance global health by implementing and sharing experience, to magnify the impact of local efforts and to pursue data knowledge improvement strategies, aiming to recognise and support best practices. EPHT can help increase the understanding of environmental public health and global health, improve comparability of risks between different areas of the world including Low and Middle-Income Countries (LMICs), enable transparency and trust among citizens, institutions and the private sector, and inform preventive decision making consistent with sustainable and healthy development. This shows how EPHT advances global health efforts by sharing recent global EPHT activities and resources with those working in this field. Experiences from the US, Europe, Asia and Australasia are outlined for operating successful tracking systems to advance global health.

## 1. Introduction

Traditionally, environmental health problems have been addressed by controlling a single pollutant or exposure. However, today’s complex environmental health problems require more innovative and holistic solutions that address not only a single pollutant or exposure, but the multifactorial effects of the environmental and environmental change on human health, and the systems that guide those effects. Considerations also need to be made at the individual, local, national and international levels. 

In fact, issues facing ‘planetary health’ [1] and the related concept of ‘ecological public health’ [2] may be the ultimate ‘wicked problems’ of our time [3]. 

According to the Lancet Planetary Health’s editor-in-chief Raffaella Bosurgi, “While public health is about health protection and health promotion within the health systems and global health looks at how to improve the health of populations worldwide, planetary health broadens this discussion by looking at the societies, civilisations and the ecosystems on which they depend. Planetary health offers an exciting opportunity to find alternative solutions for a better and more resilient future. It aims not only to investigate the effects of environmental change on human health, but also to study the political, economic, and social systems that govern those effects” [4].

Haines et al. described one holistic approach [5]: Planetary Health Watch, a proposed monitoring and forecasting system that links human health and environmental indicators in time and space, and which “…could improve the effectiveness of adaptation and mitigation strategies, assess progress towards nationally and internationally agreed targets, act as an early warning system, and hold decision-makers accountable. Indicators for inclusion in the system should be prioritised using transparent criteria, including relevance, sensitivity, sustainability, scalability, accuracy, economic viability, and consistency”.

Accordingly, information on the pollution of air, water, soil, food as well as consumer products may be used in more than one way to recognise multiple links with health, wellbeing and environmental sustainability, and support activities directed at maximising these. For this to be feasible, appropriately aggregated and linked data needs to be shared with a wide range of users, who can both contribute and gain data and interpretative frameworks consistent with their respective sphere of activity. Legal, ethical, professional and technical aspects need to be addressed for such linkages to be feasible and for access to data by users who can contribute to relevant activities.

Environmental and Public Health Tracking (EPHT) could be a concrete tool to pursue such a need [6]. EPHT can be defined as: “The ongoing collection, integration, analysis, and interpretation of data about environmental hazards, exposure to environmental hazards, human health effects potentially related to exposure to environmental hazards. It includes dissemination of information learned from these data and implementation of strategies and actions to improve and protect public health” [7].

It is an approach that helps to increase the understanding of environmental public health and global health, improve the comparability of risks between different areas of the world, enable transparency and trust among citizens, institutions and the private sector, and inform preventive decision making.

This paper aims to show how EPHT advances global health efforts by sharing recent global EPHT activities and resources with those working in this field. We describe experiences of systems in the US, Europe, Asia and Australasia and outline the components for operating successful tracking systems to advance global health. It is noteworthy that, as a whole, these systems didn’t implement new informative flows; however, first and foremost, they integrated those that were already running by integrating disciplines, institutions and professionals.

## 2. Why and How Environment and Public Health Tracking Can Help Environment and Health Integration

EPHT is also a helpful tool for strengthening the established Driving Forces, Pressures, State, Exposures, Health Effects and Actions (DPSEEA) framework [8]. EPHT promotes a systematic integration of the aforementioned DPSEEA components, taking into account both environmental and health parameters, in the context of realistic drivers, pressure and states.

EPHT aims to promote a resilient society by analysing complex datasets, addressing different audiences and supporting environmental health messaging tailored to each audience:

***The public***: information to support individual changes in attitudes and collective actions.

***Professionals and stakeholders*:** tailored information to health professionals, land planners, environmental managers and researchers.

***Decision-makers*:** integrated health and environmental information to inform decisions and create opportunities to reduce the multiplicative impacts associated with rapid urbanisation, globalization and climate/social/economic change.

Such general and generic categories also include resource managers, planners, economists, conservationists, indigenous and locally impacted communities, community developers and many other essential stakeholders. They are all strategically important, taking into account the dynamics which interrelates the two central issues on how population health may be improved: individual behaviour and social and economic factors [9,10].

The EPHT approach strives to achieve its vision of “Healthy Informed Communities” by empowering environmental and public health practitioners, healthcare providers, community members, policymakers and others to make information-driven decisions that affect health while maintaining appropriate data protection measures [11]. We now need a global perspective, demanding “new coalitions and partnerships across many different disciplines” [12]. These challenging objectives should be delivered from the perspective of providing comprehensive integration within a “planetary” framework for environmental and public health outcomes; this must be considered the ultimate goal [13,14,15].

In summary, EPHT is an instrument which can support the cross-sectoral integration of information to assist decision-making in support of the greatest ambitions for global and planetary health outcome by means a comprehensive and ecological public health prevention approach.

## 3. The Concepts

EPHT aims to merge, integrate, analyse and interpret environmental hazards, exposures and health data [16] (Figure 1) to provide information for public health decision-makers to reduce the environmental burden of disease.

Accurate and timely surveillance data permit public health authorities to determine disease impacts and trends, recognise clusters and outbreaks, identify populations and geographic areas most affected, and assess the effectiveness of public health interventions [18]. By effectively linking standardised environmental and health data in an ongoing manner, and translating it into meaningful information (Figure 2), EPHT can help to protect the health of the public.

Thus, EPHT represents a modern surveillance system, the essence of proactive public health practice, with the ultimate goal to guide public health preventive action.

Ideally, exposure tracking includes the systematic measurement of harmful environmental agents to which individuals are exposed. Exposure tracking also helps evaluate the effectiveness of public health policies by monitoring changes over time. It needs to be closely coordinated with ongoing hazard tracking. This involves the monitoring of individuals, communities or population groups for the presence of an environmental agent or its metabolites by means of Human Bio Monitoring (HBM) investigations. Exposure (and hazard) tracking is sufficient for public-health surveillance when the causal link between exposure and health effect has been established with sufficient time latency between exposure and effect in cases where the concentration-response functions are known, and where exposure measurements are representative of the population’s exposure. Examples of these situations include exposure to chemicals in drinking water [20].

The final component of environmental public health surveillance is health effects tracking, which represents traditional public health surveillance efforts. Examples of these situations are lead poisoning [21], hospital admissions for bronchiolitis [22] and some congenital malformations [23]. The key target of such a system is the primary prevention of chronic disease. Public health in the 21st century has the potential to recognise the environmental precursors of noncommunicable and communicable diseases. The case has been made in several specific instances, for example, where diabetes (Figure 3) has been associated with lifestyle factors and environmental exposure, including chronic exposure to cadmium and arsenic [24,25]; similar levels of causation apply to most noncommunicable diseases.

By highlighting the various potential levels of prevention interventions, the framework proposed for diabetes can be extended to other public health issues, as recognised by the WHO STEP-wise approach to the surveillance of noncommunicable diseases [26], where STEP established a conceptual framework that recognises the potential for prevention interventions at all levels; however, it has not yet achieved integration with the evidence base from environmental epidemiology. Therefore, improved coordination between noncommunicable and communicable disease programs with the evidence base from environmental epidemiology is called for, to put to use the available evidence on precursors of disease (i.e., environmental exposures) within a public health conceptual framework. In this way, it will be much more feasible to motivate and design the development of effective interventions at the appropriate level, to benefit from all available evidence and, thereby, to achieve the WHO goal of “Health in all policies” [27]. The two classical components of traditional surveillance include monitoring data on exposure and/or health outcomes [28].

A key distinction between EPHT and traditional surveillance is the emphasis on data integration across hazard, exposure and health information systems [3]; this could also be called “risk tracking”, which involves quantifying and monitoring, at the population level, trends in the relationship between environmental hazards, exposures and health indicators.

To ensure the success of an EPHT programme, there is a need to involve several different constituencies in public health activities. Specifically, there is a demand for timely, accessible, accurate, representative and interpretable information about our environment and health for the public, media, researchers and policymakers, including input from specific interest and community groups, as well as the public health community at large.

## 4. Some Experiences of Environmental Public Health Tracking

Tracking activities have been conducted throughout the world. In many countries, they have adopted the label ‘tracking’ (e.g., the U.S., the U.K., Australia and China), whilst in others, different labelling has been used, not necessarily calling it ‘tracking’ (e.g., France, Italy, Brazil, Canada, New Zealand) [20], even though environmental health systems contain the same components of EPHT (Table 1), but without a clear and systematic strategy to integrate hazard, exposure and health data to be properly addressed to communication. Hereafter some experiences in which authors are involved are briefly described.

In 2000, the Pew Environmental Health Commission released a report on the state of environmental public health in the United States [7]. They recommended the development of a system to track and link environmental agents, exposures and related diseases because there was a lack of basic information that could document possible links between these factors. In 2002, the National Environmental Public Health Tracking Program (Tracking Program) was created at the Center for Disease Control and Prevention. Since its inception, the Tracking Program has worked closely with a community of funded state and local health departments to build capacity and infrastructure to develop the National Environmental Public Health Tracking Network (Tracking Network), an integrated network of environmental health surveillance data at the local, state and national levels.

The Tracking Network currently provides surveillance data on 20 different environmental and health topic areas (Figure 4) and there are over 420 different environmental health measures that are publicly available. The application of these data is key to supporting evidence-based decision making and public health actions within state and local programs to help promote healthy and informed communities. For example, at the national level, Strosnider et al. [29] examined the associations between ground-level ozone and fine particulate pollution and ER visits for asthma, chronic obstructive pulmonary disease (COPD) and respiratory infections. While previous studies focused on single cities, the authors leveraged the data available via the Tracking Program to look at the association between air pollution and respiratory ER visits across hundreds of U.S. counties. At the state and local levels, there have been efforts to use tracking program resources and/or data in establishing unique and diverse partnerships, developing innovative ways to use the data and resources, and identifying approaches to making the data more accessible, all to improve public health at the local, state and national levels [30].

Meanwhile, EPHT in England [32] includes several programmes of surveillance of environmental hazards, exposures and health outcomes [33], e.g., population exposure estimation of arsenic in private water supplies [34], the burden of disease of carbon monoxide poisoning, [35] lead exposure in children [36,37], developing methods of risk prioritisation to support environmental public health interventions [38] and guidance for investigating non-infectious disease clusters from potential environmental causes. [39] The English EPHT programme has adopted an approach providing common governance for disparate themes, with the flexibility to establish surveillance structures and functions appropriate to specific information needs. Recent developments include developing national systems for enhanced air pollution exposure surveillance and weather data for public health use [33]. At a local level, the concepts of EPHT have been applied by Sandwell Metropolitan Borough Council, an urban local government in the Midlands, to identify the largest environmental public health concerns for the local authority area to help prioritise interventions [40].

In France, environmental-health surveillance has been built developing environmental-health dimensions of routine surveillance data systems. The overall concept of public-health surveillance at the national public health agency, Santé Publique France, formerly known as French Institut de Veille Sanitaire (InVS), is based on the observation that a complex changing environment creates new situations and emerging risks for which specific surveillance techniques are required (Figure 5).

Regarding specific surveillance, the European Apheis and Aphekom surveillance projects on air pollution and health [41], coordinated by Santé publique France, were designed to meet the information needs of environmental and public health institutions by performing health-impact assessments on the short- and long-term effects of air pollution over time using routine mortality and hospital admissions data. These initiatives were successful because they built on a Europe-wide collaborative network from the bottom up to stimulate cooperation and facilitate decision-making on the local and national levels [42]. In France, the surveillance programme on air pollution and health (Psas) has already celebrated its 23^rd^ anniversary, supporting French policies on air pollution at the local and national levels [43].

After the 2003 heatwave in France, Santé Public France developed SurSaUD [44]^®^, a Syndromic Surveillance system (SyS) complementing traditional specific surveillance systems, capable of detecting new threats to public health as diverse as environmental phenomena or emerging infectious diseases. The Triple-S (Syndromic Surveillance Survey Assessment towards guidelines for Europe) project, also coordinated by Santé Public France, outlined SyS activities in Europe [45]. It assessed SyS, intending to produce guidelines for human and veterinary SyS in the Member States, as well as a proposal for a European SyS strategy.

Examples of the complementarity between specific and SyS in environmental health in France include the Heat Health Watch Warning System, Cold-related diseases, Carbon Monoxide surveillance, Poisoning surveillance and Xynthia storm (2010) [46], and industrial accidents like the health monitoring of a gas leak at the Lubrizol company (2013) [47].

There are many other examples of tracking activities carried out across the world (Table 1). These countries have shared their experiences of developing EPHT and form an international network [16].

In summary, we have outlined some global examples of environmental and health information integration, which highlights the notion that factual and common constraints can be overcome.

Such reviews of tracking and similar programmes around the world indicate that EPHT has the potential to facilitate the translation of science into public health practice and go beyond just providing data or information to users and stakeholders, helping generate intelligence that is sufficiently mature to be translated into actions [30]. This recognition led to the creation of the International Network on Public Health and Environment Tracking (INPHET) in 2013 [16].

Following meetings in Europe and the USA [16], efforts have been made to identify INPHET goals aligned with broader public health needs while retaining a focus on environmental health surveillance. This network operates voluntarily, with political and intellectual independence, promoting scientific rigour in environmental and public health decision-making. Several workshop and symposiums have been held to demonstrate its application [16].

However, the EPHT tool is not yet widely available in most regions which are vulnerable to environmental hazards in the world. Some efforts are underway in some Low- and Middle-Income Countries (LIMCs) such as Georgia [63], Turkey [63], Ghana, Ethiopia, Myanmar and Tonga, to show its application and usefulness.

More generally, a detailed and comprehensive description of those experiences could be helpful to realise that it is possible to implement such organisation, but the ‘governance’ of these programmes must also be tailored according to the social, cultural and political setting. Such an issue is supported by the efforts of INPHET. 

## 5. Extending EPHT Capacity

The core infrastructure of EPHT within national public health agencies can deliver both the capacity to support ongoing concerns regarding hazardous pollutants and chemicals in drinking water, land, food and air, and new perspectives on the central value of ecological and social factors in affecting health and wellbeing in the course of multiple transitions currently experienced by society. The latter perspectives are explored initially in research partnerships between national public health agencies operating EPHT programmes and academic or other relevant research institutions. An example is the establishment of “Health Protection Research Units” in the UK, providing 5-year research programmes on topics such as Health Impact of Environmental Hazards [71] contributing methods for indoor air tracking, and Environmental Change and Health [72] addressing the role of ecological factors such as those affecting the distribution of vectors [73], climate variability affecting infectious diseases [74,75,76] or coastal changes affecting toxin-producing algae [77], as well as social factors affecting use of green spaces and related health benefits [78].

Novel interpretative frameworks could first be tested and documented in a “experimental” research setting, in studies codesigned and coproduced with public health agencies including EPHT operators [79,80]. This process facilitated appreciation of the value of the mentioned alternative interpretative frameworks as components of the routine operations of public health agencies including provision of advice and EPHT. For some topics, the usual delay between research and practice could be considerably shortened by these arrangements. This type of collaboration illustrates the capacity for EPHT to act as a fast transmission chain from existing information frameworks to their transformation into public health tools capable of systematic consideration and integration of ecological and social factors. This process means that EPHT may support mainstream public health operations and partnerships in making a transition toward more appropriate consideration of ecological and social factors in health protection and health improvement activities.

The process briefly summarised here for the case of EPHT in England has been applied in different forms to other settings and countries.

While the specific modalities vary in terms of form and amount of research fund allocation and its relationship with EPHT, a common pattern is emerging where EPHT connects with a range of research approaches and results, and aims to integrate them into everyday environmental health activities.

## 6. The Future

The continued development of EPHT activities around the world will help support improvements in environmental health. In aligning with the WHO Health in All Policies (HiAP) strategy [81], EPHT can provide a beneficial approach to public health practice in an era of diminishing resources and increasing demand, where scientific evidence can support interventions in a range of sectors, such as transportation, housing, energy, waste management, land use and climate change. However, action at international, national, regional and local levels must be taken to improve and protect global health as soon as knowledge and intelligence become appropriately mature to be able to effectively address, monitor and evaluate progress against public health risks. Many actions to improve the environment can have multiple health co-benefits, such as policies promoting active travel in cities which can reduce air pollution and noise from vehicle traffic and, at the same time, help improve physical activity and respiratory and cardiovascular health [82].

EPHT will also promote strong and enduring relationships and partnerships across all tiers of the local and national government, across public health and environmental agencies, and with both the private sector and the general public/NGO community. Along this line, the focus should be placed on the issues described in Section 5.

## 7. Governance: Principles to Direct EPHT

The collective experience of EPHT operators in several countries points to several fundamental principles for the good governance of such an operation. Such principles have developed over decades, and therefore, may not appear innovative; however, they are valid based on a widely experienced practice.

***Sustainability***: This principle refers to the need for any surveillance/tracking system to produce useful outputs on an ongoing basis. Such a need is founded on minimal human resources, skills and capability to be maintained. It also relates to the sustainability of inputs of data feeds and the ability to produce regular outputs of surveillance, such as annual reports.

***Competence***: In principle, any public health surveillance/tracking system requires leadership by a public health agency, operation by staff trained in epidemiology and public health information systems.

***Integration*:** A worthwhile goal for an EPHT system is to contribute to addressing the overall preventable burden of disease. Starting from a topic within this broad agenda will require integration of complex environmental, exposure and health information.

***Accountability*:** An EPHT has the potential to produce influential information for decision-makers and officials in a range of public agencies. Typically, this will require a multidisciplinary collaboration where competencies from several departments or organisations are contributing to the same agreed surveillance/tracking goal. For these reasons, accountability via a clear and an agreed governance approach is necessary (i.e., through appropriate Terms of References).

***Transparency*:** Specific tasks of members of a surveillance/tracking working group can be allocated by a mechanism agreed within the working group itself and in consultation with general policies and practices. This will typically involve adopting a governance tool for recognising the task and role of the group.

In other words, governance should essentially aim to federate data providers and standardise the methods of data collection and analysis at the level of each established partnership. The above governance principles are largely consistent with dimensions recognised by OCAP principles (ownership, control, access, and possession) [83].

## 8. Ethical Dimensions

The four principles of beneficence, nonmaleficence, justice and respect for autonomy are included in ethics guidelines drafted for public health professionals. They are particularly relevant in the field of EPHT because any activity which is capable of changing the extent of emissions or pollution necessarily requires a careful balancing of several contrasting considerations in the interest of optimally addressing the needs of society. Although the following do not provide an exhaustive account of how the principles are applied, they can be used as a framework to support ethical decision making.

### 8.1. Beneficence

#### 8.1.1. Confidentiality and Privacy

Maintaining the confidentiality of personally identifiable data (PID) is a key feature of any research study, and is also relevant to the management of data assets part of a public health surveillance programme such as EPHT. It is important that participants are treated with respect and dignity at all times [84], although EPHT mostly will benefit from assembling data assets from data collected for other reasons, thus not adding a burden to participants while being careful to preserve anonymity.

#### 8.1.2. Data Storage and Data Sharing. 

Hard (quantitative) and soft (intelligence) data [85] need to be stored securely, should be anonymised and archived for a period appropriate to the purpose; as EPHT focuses on noncommunicable disease, holding aggregated data over decades would be necessary for comparisons over time (20–30 years).

### 8.2. Nonmaleficence.

#### 8.2.1. Minimise Risk, Disruption and Harm

Nonmaleficence is achieved by assembling data assets for an EPHT programme that minimises risk, disruption and harm to both study participants and the source population. These risks should be fully assessed, quantified and mitigated or minimised if possible. If a risk is discovered as part of EPHT-based analyses that might adversely affect well-being, it should be communicated to the individuals/populations concerned [84].

#### 8.2.2. Precautionary Principle

Another example of nonmaleficence is the ‘Precautionary Principle’. This essentially means that “When human activities may lead to harm that is scientifically plausible but uncertain, actions shall be taken to avoid or diminish that harm [84,86,87], though not advocated by all” [88].

### 8.3. Autonomy (Including Informed Consent).

#### 8.3.1. The Distinction between Data for Research and Public Health Surveillance

An important consideration when applying for ethics approval is to establish the status of the EPHT programme and data assets. Accurate definitions may be necessary to discern the level of ethical approval required [89]. 

#### 8.3.2. Institutional Review Boards/Research Ethics Boards 

There are numerous names for groups of people officially established to decide whether a research project can ethically and legally be approved, such as Institutional Review Boards, Research Ethics Boards, and Research Ethics Committees (RECs). Their constitution may differ across time and jurisdictions, with various experts brought in if required by the type of study being assessed [90], but they should always contain representatives of the population involved [84].

#### 8.3.3. Informed Consent

The International Society of Environmental Epidemiology (ISEE) Guidelines state that there must be full disclosure of relevant aspects of the study, such as its purpose and any potential hazards, to the study participants before explicit prior, documented informed consent is obtained. [84].

### 8.4. Justice

#### 8.4.1. EPHT Obligations to Society

Obligations to society involve ensuring objectivity in the process of designing of an EPHT data asset and reporting of the outcome of specific analyses, as well as the overarching obligation to deliver only high-quality public health surveillance, supported by relevant research which will materially improve the understanding of a particular subject.

#### 8.4.2. EPHT Programme Obligations to Funders/Sponsors, Employers and Colleagues

These obligations need to be identified early in the research process and involve the population concerned at all stages of the study, from initiation to publication [84]

## 9. Examples of Priorities for EPHT Activities

Taking into account what can be reasonably and practically implemented in the near future, and the needs of many countries in particular in LMICs, we suggest some issues to which efforts should be devoted. They could be some realistic field in which to apply the ‘holistic’ framework of EPHT because of a new and emerging environmental health challenge relevant to planetary health.

### 9.1. EPHT in Urban City Planning

Evidence-based interventions to develop an integrated approach to improve air quality and climate change adaptation in cities are being implemented. EPHT can help evaluate urban policies to combat extreme temperatures, air pollution, noise, promote green spaces, sustainable urban development, etc. as urged for example by the WHO [91], Climate Clean Air Coalition’s Urban Health Initiative [92] and Healthy Polis Initiative [93].

### 9.2. EPHT in Industrially Contaminated Sites

Development of industry and its products has brought many benefits to modern societies, including a reduction in deprivation. Conversely, it has also generated a large amount of hazardous materials, in many cases where some of the most disadvantaged and vulnerable communities live. Such activities may harm health and wellbeing due to chemical exposure and socio-economic deprivation [94]. Industrially contaminated sites (ICS) represent a long-term legacy of past and current development, a probable lasting cause of preventable noncommunicable disease and a living reminder of the inherent lack of sustainability of the linear economy. There is an urgent need to identify the most suitable interventions aimed at prevention of ill-health in affected communities, to facilitate better social and economic development while minimising exposure to harmful compounds associated with ICS.

### 9.3. EPHT and Socio-Economic Development

In LMICs, there is a strong need for economic development. Historical patterns worked well economically for the high-income world but resulted in burdening the environment and health across the globe and in particular for LMIC. The solution requires a technological and intellectual transition in all countries; an alternative route to economic prosperity that preserves resources and limits carbon emissions is urgently needed. EPHT can be a helpful tool with which to work towards informed, healthy, sustainable and equitable prosperity in these developing economies.

## 10. Conclusions

Environmental public health issues are becoming increasingly complex. Globalisation, population growth and overconsumption are placing significant stresses on the environment and health.

Socioeconomic diversity across the world must be taken into account, which is closely related to a geographic diversity in data/information availability and ability to use these resources to inform decision-making.

EPHT can contribute to reducing socioeconomic and environmental inequality across communities, countries and regions, by sharing experiences, knowledge, information and data. Hence, EPHT networking activities must support local, regional, national and global improvements in the environment and reductions in its impact on health, which will be achieved by strengthening and sharing a common philosophy among public health professionals working in environmental health around the world. It requires governance efforts for the integration of a wide range of scientific disciplines and people, from institutional decision-makers and officials in public health agencies to representatives of the civil society. In sum, this article:Indicates that difficulties encountered in integrating disciplines, professional profiles and institution can be effectively overcome by focusing on the final goal of environmental health prevention and promotion. Some successful stories are herein described in different countries, either under the banner of EPHT or not, but with the same function;Introduces the experience of countries, where such an approach is being implemented nationally, whilst taking the opportunity to collaborate with other countries (hubs);Focuses on some lessons learned in performing such experiences, first and foremost from the ethical point of view;Discusses the concept of EPHT, aiming at sharing opportunities to develop it towards more ambitious goals (planetary health);Clarifies the context in which such an ecological public heath tool might be essential to attaining global health goal, but also to moving towards the ultimate goal of Planetary Health.In other words, it aims at creating a community of health professionals and researchers who could share experiences, proposals, and thoughts in the field of practical, effective environmental health prevention and promotion.

## Figures and Tables

**Figure 1 ijerph-17-01976-f001:**
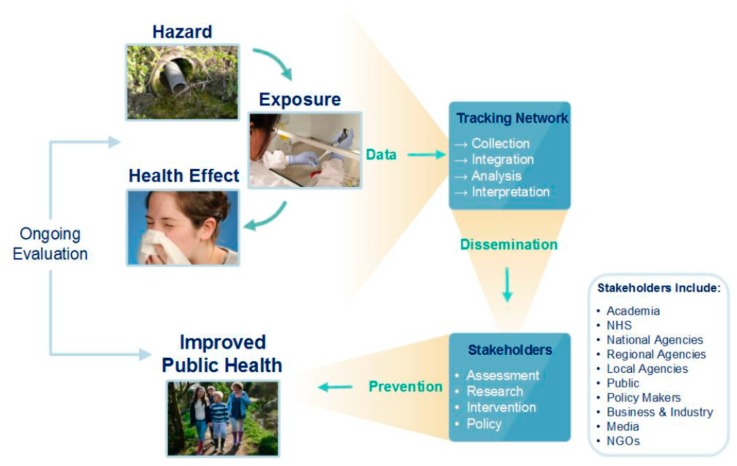
Environmental Public Health Tracking Components [17].

**Figure 2 ijerph-17-01976-f002:**
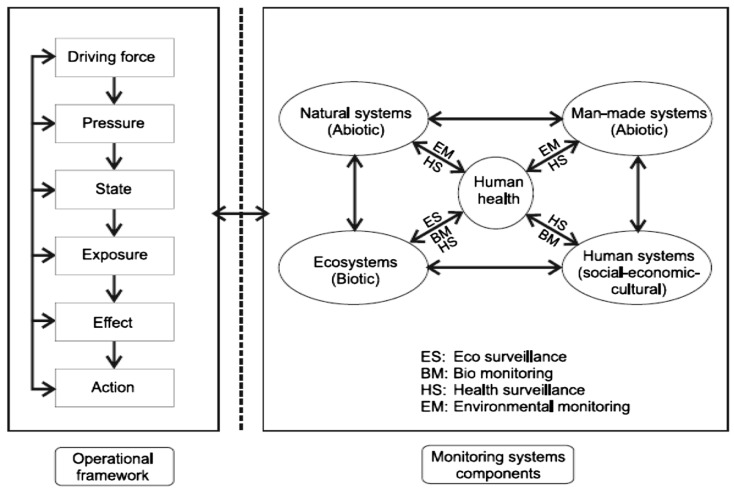
A conceptual framework for integrated environmental health monitoring [19].

**Figure 3 ijerph-17-01976-f003:**
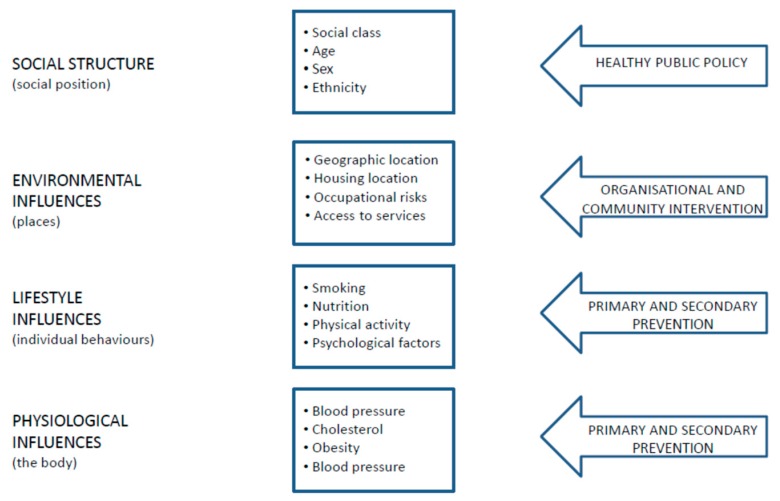
Levels of causation and corresponding types of intervention [26].

**Figure 4 ijerph-17-01976-f004:**
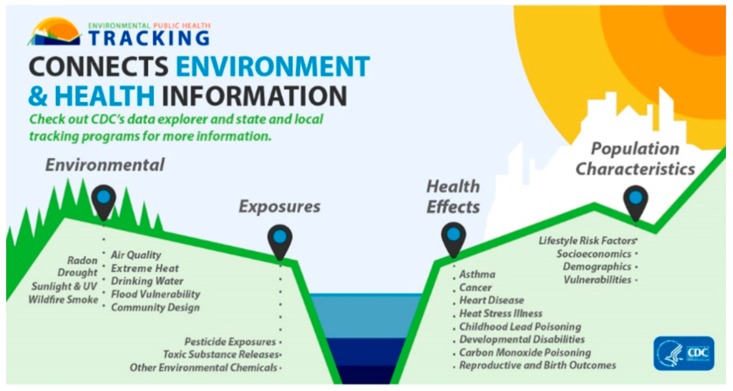
Environmental Health topics investigated by the US EPHT [31].

**Figure 5 ijerph-17-01976-f005:**
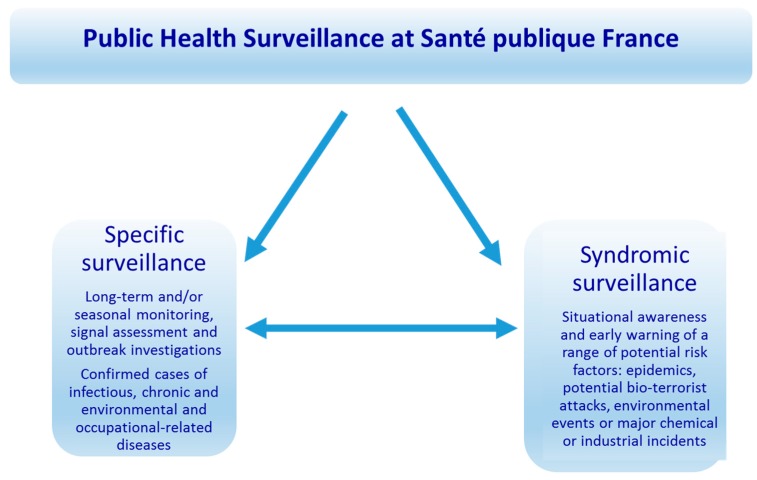
Public Health Surveillance at Santé publique France.

**Table 1 ijerph-17-01976-t001:** Tracking activities around the globe that are collaborating in an international EPHT network (INPHET) [16].

Country	Name/ Subjects	Leading Institution
**EPHT SYSTEMS**		
**Australia** [[48]	Currently developing a strategy for an Environmental Health Tracking System in Victoria. A Driving Force-Pressure-Environmental Condition-Health Impact-Action (DPEHA) conceptual framework is proposed for the proposed Victorian EHTS.	Environmental Protection Agency (EPA) Victoria’s Environmental Public Health Unit
**China** [49,50,51].	The Chinese Environmental Public Health Tracking (CEPHT) project started in September 2015 operated by the National Institute of Environmental Health, Chinese Center for Disease Control and Prevention (NIEH, China CDC), developing CEPHT’s electronic data tracking system. Twenty-nine local Chinese CDCs participate by reporting environmental hazard and health effects data through the electronic system.	National Institute of Environmental Health (NIEH, China CDC)
**EPHT-EQUIVALENT SYSTEMS**		
***North America***		
**Canada,**	Environmental Public Health Program: Focus on outdoor air quality, water quality and soil contaminants. Health databases include emergency department visits, hospital admissions, cancer and mortality data [52] Acute Care Enhanced Surveillance (ACES) is a real-time syndromic surveillance system with temporal and spatial capabilities that enables public health to be better informed on the health of the community. ACES’ syndromic surveillance capabilities are useful in a variety of situations, including public health emergencies, such as extreme weather events. [53]	Canadian Urban Environmental Health Research Consortium (CANUE) The ACES system is maintained by KFL&A Public Health and is funded by the Ministry of Health and Long Term Care
***Europe***		
**European Union**	European Environment and Epidemiology (E3) Network [54], providing access to climatic/environmental geospatial data for epidemiologic analysis currently collected and analysed by a variety of European agencies, public health institutes, and research organisations The system is an interactive database composed of country-level indicators and regional assessments [55]. ENHIS indicators provide information on exposures, health outcomes and policy actions related to the environment and health priority areas for the European Region known as Regional Priority Goals (RPGs) The initiative is coordinating and advancing human biomonitoring in Europe. HBM4EU is generating evidence of the actual exposure of citizens to chemicals and the possible health effects to support policymaking [56].	European Centre for Disease Prevention and Control (ECDC) *Environment and Health Information System* (ENHIS) *European Human Biomonitoring Initiative* (HBM4EU)
**Italy**	The main activity that has characteristics of ongoing surveillance is on “Mesothelioma and Asbestos exposure tracking” [57]. There are also several studies with the potential to become ongoing surveillance programmes. Sentieri [58], which describes the health profile of populations living in contaminated sites, and provides elements for the design of ongoing monitoring, that could have value across several countries [59,60] Moniter [61], dealing with incinerator risks EpiAir [62] for air pollution surveillance Environmental Health Task Force (EHTF) appointed by the Ministry of Health (MoH), is underpinning a strategy to develop a framework which includes local and national resources (underway).	National Institute of Health National Institute of Health ARPA-ER DEP-ARPAP EHTTF-MoH
**The Netherlands**	Small Area Health Studies using health registries [63] Statistics Netherlands (CBS, Ministry Economic Affaires), Strategic Project Health Care [63] Lung cancer in the IJmond region in relation to cadmium exposure [64] Aircraft noise annoyance in the vicinity of Amsterdam Schiphol Airport [65]	National Institute of Public Health and the Environment (RIVM)
***Eurasia***		
**Georgia** [63]	A Multiple Indicator Cluster Surveys (MICS) on a representative sample of children to study several demographic and health aspects over the last ten years. Starting from lead biomonitoring in children, a national surveillance programme is underway.Considering the geographical and geopolitical situation of the region, the experience in Georgia may lead to an initiative in collaboration with neighbouring countries, with the potential to establish a regional Eurasia hub for EPHT, based on a network established at a recent EU funded TAIEX workshop.	National Center for Disease Control and Public Health (NCDC-PH)
**Pacific Region**		
**New Zealand**	A system reports on a range of environmental hazards, with the additional synthesis of the environmental burden of disease for second-hand smoke exposure. Also working on developing environmental burden of disease reports for ultraviolet light exposure and lead (Pb) exposure. [66,67] Reporting provides weekly communicable disease surveillance and outbreak surveillance reporting to New Zealand regional health authorities.	Massey University Institute of Environmental Science and Research [68]
**Kingdom of Tonga** [69]	Vector and vector-borne disease surveillance that includes meteorological and Southern Ocean oscillation index data, and relates to the climate change monitoring data collected in Tonga Water quality, water-borne disease and infection monitoring	Ministry of Health, Kingdom of Tonga
**South Pacific Region** [70]	The Pacific Public Health Surveillance Network (PPHSN) is a voluntary network of countries and organisations dedicated to the promotion of public health surveillance and appropriate response to the health challenges of 22 Pacific Island countries and territories (PICTs). It includes six service networks: PacNet, LabNet, EpiNet, PICNet, the Pacific Syndromic Surveillance System and the Strengthening Health Interventions in the Pacific-Data for Decision Making (SHIP-DDM) capacity development programme.	Secretariat for the Pacific Community (SPC)

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
