# Peer review of "Advancing Global Health through Environmental and Public Health Tracking"

_ijerph, 2020, doi:10.3390/ijerph17061976_

Round 1

Reviewer 1 Report

This paper attempts to build the idea of EPHT as an innovative response in the wake of a planetary health crisis. The paper suffers from some fairly major flaws, but the most glaring is the mismatch between the planetary health framing, EPHT and the examples provided. For example, your overviews of US and English EPHTs all draw from VERY classical environmental exposures with 50 year research histories. I’m not convinced that any of this is new, or necessarily engages with planetary dilemmas outlined and articulated in the introduction. I think the paper would be an easier sell as a review of environmental tracking. It’s trying too hard to be ‘new’ when the information represents a very classic epidemiological orientation. Moreover, you don’t articulate how any of these activities align with either an EPHT framework, or DPSEEA limiting the effectiveness of argumentation and minimizing a value-add of your paper. Additional issues are described below:

P1L40-41 - awkward transition. I think you need a linking sentence moving the reader from planetary health to assessment methods

p2 l3 0 could be good to briefly introduce how EPHT links with the holistic ideas presented by haines et al?

P2l13 - you have not adequately introduced EPHT for it to resonate with DPSEAA. what is it and how are actions accounted for in this monitoring framework? Move the definition from a footnote to the text for reader

L31 - it’s not clear how EPHT empowers communities brother groups? what mechanisms are built in to support empowerment, capacity-building, training and evaluation? How do the case studies you reference do these activities? Right now it just reads as any old information system rather than a comprehensive suite of tools.

L39-40 - clarify which goals, specifically…

Fig 1 - it’s not clear from this figure how EHPT differs from traditional surveillance approaches or is supportive of DPSEAA/planetary health. This is a more or less conventional surveillance diagram. It’s also not clear from your articulation above how EPHT engages in evaluative activities. Moreover, the text that follows doesn’t say anything about time demand. Is this meant to be real-time surveillance?

P5L3 - how can a monitoring system be proactive when it explicitly is designed to detect events in the past?

P5L9 - run on sentence. lacks clarity.

P6 L4 - you need to specify that figure 3 is a diabetes specific framework

P6L11 - how does your thinking apply to healthy public policy(e.g.health in policies outside the health sector) when the entire paragraph speaks almost exclusively to health sector actions?

P6L16-19 - This reviewer questions how novel this is, really. Many surveys and data sets incorporate this kind of information to controller multiple exposures. It’s still not clear how EPHT is different from traditional surveillance, which also includes tracking information about known risks or integrating other data sets.

L25 - you need to contextualize your WHO quote in relation to the kind of information EPHT is designed to track and communicate, and how it is communicated (e.g. OCAP principles)

L31-32 - you indicate that environmental tracking systems in these countries are similar to EPHT… so again… what is novel about what you are advocating for??

P9- Table - Canada has many other surveillance systems including ACES—a syndromic surveillance system used for heat. CANUE is a national research project with 5 years of funding to develop a data repository. Beyond providing data from regularly collected data sources such as the census, satellite data, etc. (which CANUE does NOT collect themselves), they don’t provide capacity building or evaluative actions or empowerment indicated above in the article which calls into question how well any of these examples align with planetary health and a ‘complex/innovaitve’ focus for EPHT. It also raises the question of what elsethis review table is missing among the listed countries, especially given the title of the table implies a comprehensive global review.
P10 L1-4 - Have you outlined how these are successful? How were they evaluated? To what ends? How do they align with DPSEEA? How do they pick up on planetary dynamics such as climate change or multiple exposures from 100 year histories of multiple industrial land use? I find the examples utilized far too simplistic given the ‘complex’ framing of EPHT in the intro

P11L6-P12L34 - have the authors not heard of OCAP principles or free/prior informed consent? Why recast the data governance wheel when good guidance already exists? It’s not clear what this adds when EPHT is already underspecified in the current draft.

P12-13 There are SIGNIFICANT missed opportunities in your future directions. If EPHT is as good as you say it is (this still needs to be made convincing to readers), why not focus its efforts to issues of planetary health relevance? Global climate change, ocean acidification/warming and the collapse of fish stocks, undernutrition related to changing climate, etc.??

P13L33-44 - I’m not convinced these goals have been met and you overreach in your statements of what the paper does and does not do. Suggest toning down language and revisiting your goals for the paper and what follows in the text. Your last line is also not in keeping with your objectives. How does your paper build a community of EPHT practitioners?

Author Response

In the coverletter

Reviewer 2 Report

I don't have further comments.

Author Response

In the coverletter

Reviewer 3 Report

Authors followed the major remarks I reised in my review,

They have expanded the article to include more details on case studies  They also added examples with more details describing tracking in different countries, such as  USA, UK and France.

I do not have any more comments.

Author Response

In the coverletter.

This manuscript is a resubmission of an earlier submission. The following is a list of the peer review reports and author responses from that submission.

Round 1

Reviewer 1 Report

Paper is presenting a concept of Environmental and Public Health Tracking (EPHT)  system at its application to very ambition  goal of ‘Advancing Planetary Health’. .

The concept of EPHT aims to merge, integrate, analyse and interpret environmental hazards, exposures and health data to provide information for public health decision-makers to reduce the environmental burden of disease.

The proposed concept is an modern version of surveillance system, the well-known public health practice, with the ultimate goal to  guide public health preventive action. The added value is that is emphasis on identification (risk tracking) of hazards in order to control exposure.

However, authors make it very clear that the key distinction between EPHT and traditional surveillance is the emphasis on data integration across hazard, exposure and health information systems.

Authors emphasis that in order to ensure the success of an EPHT programme, there is a need to involve several different constituencies in public health activities. Specifically, there is a demand for timely, accessible, accurate, representative and interpretable information about our environment and health for the public, media, researchers and policymakers, including input from specific interest and community groups, as well as the public health community at large.

IT seems to be a real challenge for implementation, especially in countries with less developed electronic data storage and IT capabilities.

Experience of environmental public health tracking across the world are shortly presented.  In my opinion it would be of the benefit of reader to provide one more detailed experience with EPHT with identification of major limitations and the information how they were overcame.

The use of “planetary” concept seems to be to ambitious, the focus on the global health would be sufficient.

Reviewer 2 Report

The concept of this manuscript is easy to follow. To establish an EPHI will benefit many aspects of the world, including improved monitoring of environmental hazards and protected target population. However, this manuscript is too brief, and there are many places that need more details. Specific suggestions are followed:

Overall, the author should clarify the purpose of this paper. Does this paper propose a global EPHI? If that, more details should be included (e.g., host organization, implementation plan). Only introducing the timeline of EPHI in different countries is not enough to show the benefits of having an EPHI system. I suggest to include more specific achievements in the “Experience” section. Line 98-213. The author mentioned that EPHI could also improve the monitoring of human diseases. I suggest specifying how to link the tracking of environmental compounds to human diseases. Line 239. What is the meaning of “hard and soft data”?

Reviewer 3 Report

This manuscript seeks to advance the fledgling growth of planetary health through bolstering of environmental and public health tracking systems around the world. While this reviewer is sympathetic to the goal of this paper and was quite excited about the framing of the title and abstract--ultimately, the paper falls far too short of the lofty goals stated at the outset and suffers from several major and minor flaws that limit its contribution and scientific rigour.

Major issues

The biggest issue with this paper is that it lays out in the introduction the need for more holistic approaches to environmental and public health tracking in the face of increasingly complex public health challenges associated with global environmental change. However, the paper rarely returns to this and the framework presented is merely a repackaging of more or less traditional epidemiological surveillance with modest integration of the DPSEEA framework. Related to the above, the authors provide no definition for EPHT (see line 76) when it is introduced. What does this include? What does it not include? On line 105 you claim this would be a 'modern' approach to surveillance, but do little to qualify this statement. On L110, you indicate the biggest difference to be incorporation of risk tracking, but this is a highly reductionist orientation to 'classical' epidemiological approacjhes for understanding host, vector interactions in manifesting health outcomes. There is nothing new or 'modern' about this orientation to surveillance, and it belies the fact that many systems around the world have been doing this type of work for decades. Perhaps teh point is this is more relevant to low-income or low-resource countries? When you introduce EPHT goals on L85, Baum and Fisher (2014) Sociology of Healht and Illness provided a good review of why behaviour change continues to fall short as an intervention to improve population heatlh. Your conclusion here therefore overreaches. Similarly for stakeholders, why such a short list for the need for a truly integrative tool? You list a few others in your dicussion section adn in the figure that follows, but you have overlooked resource managers, planners, economists, conservationists, indigenous and locally impacted communities, community developers, etc to name but a few relevant 'stakeholders;. For L88 - there has been much written about the need for integration in health and environment interactions. See Buse et al. 2018 J of Epi and Community Health; Gillingham et al. 2016 Integration Imperative; Parkes et al. 2019 Challenges; Buse et al. 2018 International Journal of Public Health Similar to above, L91 overreaches as you have not provided sufficient evidence to support the core argument that more disease tracking solves population health challenges. This is awfully reductionist and misses the complexity of approach for true intersectoral action on these issues that an 'integrated approach' would support. Thus, on L95, you have yet to really introduce anything 'new'? Moreover, this statement seems to suggest that orgs like the WHO or UN have not maintained a 'global perspective' of these and related issues which is patently untrue. Figure 1 is awfully reductionist and a simple recasting of just about any surveillance system. This is problematic given your critique of those systems in the opening lines of the paper. Moreover, it misses some of the fundamental governance principles that need to be espoused by the system which you identify in your discussion. Even then, none of those governance principles are particularly new or innovative in relation to data collection adn surveillance. L124 - given that countries call these systems by different names, perhaps you could provide the reader with an overview of different framings, otherwise we can't fact check what you have looked for and where you have looked in order to understand how you are conceptualizing these systems and how they manifest in certain parts of the world. The section titled Experience of Environmental PH Tracking Across the world is an innappropriate title. This is not globally comprehensive in any meaningful way (missing the middle east, Africa, SE asia, and many other parts of the world.) Moreover, you have equated single country experiences to entire continents (e.g. North America is comprised of more countries than just the USA, and differences between local, state and fedearl systems in Mexico and Canada and the US all very in their degree of integration). This critique is salient of every heading in this section (e.g. Europe is more than just the European CDC and PH systems in England, Italy and France). The fact that this review seems incomplete, unstructured and not at all rigorous considerably diminishes the contribution of this work. It would have been better for you to take a case study approach and look at 3-5 PH systems that you think ARE doing a good integrative job that align with your framework and use them to establish what's working and why its working. Otherwise, there is no selection criteria for what you are looking at in this paper or how it relates to your core argumentation around need and scope for EHT. Relatedly, you indicate in your intro that you want to provdie a review of these experiences, but in L182 you indicate that a detailed and comprehensive description could be helpful... You haven't provided either of these and therefore the goal in the intro is overstated. Again, related to the figure, it would be good to see the princi0ples of good governance built into the framework in Fig 1. However, it's not clear how an improved surveillance system will magically form new intersectoral relationships (L201)--and much has been written about these challenges in the literature which are uncited in the paper--which oversimplifies the relationship between PH tools, and PH processes to take knowledge and use it to drive action. Thus, your EPHT is a black box for the reader--you claim it can do all these things, but provide no evidence or rationale or pathway by which all the good it will supposedly produce will manifest... Moreover, and related to the above, for a system that is supposed to be integrative, the idea of integration here is woefully limited to environment/health interactions. You overrely on quantitative methods for tracking (another form of integration) and say nothing about the time and spatial challenges present for complex environmental health challenges. On L277, you have a considerable opportunity to apply your 'holistic' framework to a NEW and EMERGING environmental health challenge relevant to planetary health... and you select contaminated sites... one of the most researched areas of traditional environmental health practice. Selecting a better case study would help demonstrate how you would achieve the integration you strive for and how the framework will actually drive better decision-making, collaboration, etc. in a holistic fashion, and which are attentive to scalar health issues present in the planetary health paradigm. given the above, the conclusion considerably overreaches the established potential of EPHT

Minor Issues

Minor grammatical errors and punctuation errors throughout. Missing key references on intersectoral action MIssing key arguments and evidence for linking new holistic tools to action (surveillance systems track, they don't drive action... so what happens after we establish relationships in the data??)